# Computational investigation of hysteresis and phase equilibria of n-alkanes in a metal-organic framework with both micropores and mesopores

Zhao Li [1], Jake Turner[1] & Randall Q. Snurr [1✉]

Adsorption hysteresis is a phenomenon related to phase transitions that can impact applications such as gas storage and separations in porous materials. Computational approaches can greatly facilitate the understanding of phase transitions and phase equilibria in porous materials. In this work, adsorption isotherms for methane, ethane, propane, and n-hexane were calculated from atomistic grand canonical Monte Carlo (GCMC) simulations in a metal-organic framework having both micropores and mesopores to better understand hysteresis and phase equilibria between connected pores of different size and the external bulk fluid. At low temperatures, the calculated isotherms exhibit sharp steps accompanied by hysteresis. As a complementary simulation method, canonical (NVT) ensemble simulations with Widom test particle insertions are demonstrated to provide additional information about these systems. The NVT+Widom simulations provide the full van der Waals loop associated with the sharp steps and hysteresis, including the locations of the spinodal points and points within the metastable and unstable regions that are inaccessible to GCMC simulations. The simulations provide molecular-level insight into pore filling and equilibria between high- and low-density states within individual pores. The effect of framework flexibility on adsorption hysteresis is also investigated for methane in IRMOF-1.

[1] Department of Chemical and Biological Engineering, Northwestern University, 2145 Sheridan Road, Evanston, IL 60208, USA. ✉email: snurr@northwestern.edu

Phase transitions and phase equilibria have been an important topic in fundamental research for decades and play an important role in many technologies. From the development of the van der Waals (vdW) equation of state[1] to the recent discovery of a secondary liquid phase of water[2], this field has experienced major advances and introduced new concepts and theoretical tools into science and engineering[3]. As an important type of phase equilibrium, adsorption of confined fluids in porous materials plays an important role in applications such as adsorption cooling[4,5], hydrogen and methane storage[6–8], and oil extraction[9]. One aspect of phase transitions and phase equilibria in porous materials where major questions remain is the phenomenon of adsorption hysteresis, where the adsorption and desorption branches of the adsorption isotherm do not trace one another.

Adsorption hysteresis and the related phenomena of condensation and evaporation in porous media have been investigated by several groups, especially using molecular modeling and theory. Gubbins et al.[10,11] used grand canonical Monte Carlo[12] (GCMC) simulations and classical density functional theory (DFT) to study hysteresis in slit and cylindrical pores and developed theories[13] that can explain this complicated phenomenon. For example, based on GCMC simulations in cylindrical pores, Coasne et al. proposed the concept of a critical hysteresis temperature, $T_{cc}$, above which hysteresis disappears. In their theory, $T_{cc} = 2\sigma T_c/R_0$, where $\sigma$ is the kinetic diameter of the fluid particle, $T_c$ is the bulk critical temperature of the confined fluid, and $R_0$ is the pore radius[14]. The equation can be rearranged to calculate a critical hysteresis pore diameter at a given temperature, below which hysteresis should not be observed. Striolo et al.[15] studied hysteresis loops of water in carbon nanotubes with various radii at different temperatures using GCMC, and they found that the size of the hysteresis loop decreases and the transition pressure for the condensation increases as temperature increases. They also found that the critical temperature of water in nanotubes decreases as the diameter of the nanotube decreases. Neimark et al.[16,17] developed the gauge cell Monte Carlo method that uses a "gauge cell" to limit the density fluctuations when the simulation is performed in the metastable and unstable region. By limiting the density fluctuations, they were able to maintain metastable and unstable states in a simulation and gather statistics about these states and form a vdW loop. Recently, they extended their gauge cell MC method to the calculation of chemical potentials of chain molecules[18] and applied it to the verification of hysteresis loops in mesopores in MCM-41 and CMK-3[19]. Ma et al.[20] applied the gauge cell method to methane adsorption in different IRMOFs[21] to investigate the impact of confinement on thermodynamic properties such as the critical properties of the fluid compared to the bulk phase. They also found through simulations that the hysteresis loops for methane in IRMOFs are affected by temperature, pore size, and strength of the adsorption energy. Höft and Horbach[22] applied umbrella sampling for methane adsorption in IRMOF-1[23] at low temperatures. Using this free energy method, they found that at low temperatures, namely from 90 K to 120 K, there are two types of phase transitions of methane in IRMOF-1. At low densities, methane exhibits a phase transition on the MOF surface, while at higher pressures there is a condensation transition.

Do et al.[24] developed a kinetic Monte Carlo (kMC) scheme in the grand canonical ensemble and used it along with canonical kMC to simulate argon adsorption both on a graphite surface and in graphite slit pores with spacings of 2 nm and 3 nm. Sarkisov and Monson[25,26] used lattice DFT and grand canonical molecular dynamics (GCMD) in disordered porous materials to imitate the experimental process of adsorption and desorption by incorporating mass transfer. They divided a simulation cell into a "control volume" region, which serves as a source or sink of adsorbate molecules via GCMC moves, and a porous material region, where the adsorbate molecules diffuse through canonical molecular dynamics. Their work shows that GCMC and GCMD simulations can reach the metastable region reproducibly, suggesting that there is a physical connection between the hysteresis loops generated by GCMC simulations and those observed in experiment. In another paper on the topic, Schoen et al.[27] suggest that hysteresis in GCMC simulations comes from the inability of the algorithm to sample the phase space ergodically within reasonable computational resources.

Most previous studies of adsorption hysteresis focused on idealized pores such as slit[11,15,28–30], cylindrical[31–36], and spherical pores[37]. These idealized models facilitate characterization of the structure/hysteresis relationships and make it easier to develop theories. Efforts have also been made to investigate the phase behavior of adsorbate molecules in complex confinements. For example, Monson and co-workers[26,38–47] focused on adsorption hysteresis in disordered porous materials such as Vycor glasses. They applied mean-field approximations and used idealized cylinders of different sizes to mimic the pore shape of porous materials with disordered cylindrical pore[38].

Recently, metal-organic frameworks (MOFs), a new class of porous materials, have been intensively studied for adsorption applications, and hysteresis is sometimes observed. For example, Struckhoff et al.[48] investigated the adsorption hysteresis of argon in MIL-101 (Cr) using both adsorption experiments and molecular simulation. They found that the two steps in the isotherm correspond to the filling of the micro- and mesopore of the structure. Hysteresis loops for $CO_2$ adsorption in open-channel MOFs were also found by Bezuidenhout et al.[49] They concluded that the strong carbon-framework electrostatic interaction is the key to the hysteresis loop at 298 K.

In this work, we studied adsorption hysteresis of methane, ethane, propane, and n-hexane in a MOF structure with a micropore and a mesopore using GCMC simulations to investigate the behavior of adsorbates in a complex pore architecture and to better understand hysteresis and phase equilibria between connected pores of different size and the external bulk fluid. We also demonstrate the use of canonical ensemble simulations with Widom test particle insertions as a complementary method to GCMC. By constraining the number of molecules in the canonical ensemble and calculating the chemical potential or fugacity using Widom insertions, we can find the stability limits and validate the hysteresis loops observed in GCMC. The Widom insertions also provide insights about the configurations of molecules in the unstable states that are impossible to capture using GCMC. For the selected system, we show how adsorption hysteresis changes with temperature and with the alkane chain length. We also analyze the heat of adsorption and its guest-guest and host-guest components and their relation to adsorbate siting and the steps in the adsorption isotherms. We also show interesting system size effects on the hysteresis loops and discuss connections between molecular simulations on (necessarily) small systems and macroscopic experimental systems. Finally, we show that Widom insertions can be coupled with molecular dynamics simulations, and briefly show how framework flexibility affects adsorbate phase transitions and condensation in IRMOF-1[23] as an example.

## Results

For our investigation, we selected MOF #667 from the ToBaCCo 1.0 database[50] because it has a very high void fraction, high surface area, and relatively large pores that may result in adsorption hysteresis. In addition, the connection between the micro- and mesopore may lead to complicated adsorption behaviors. MOF #667 has two types

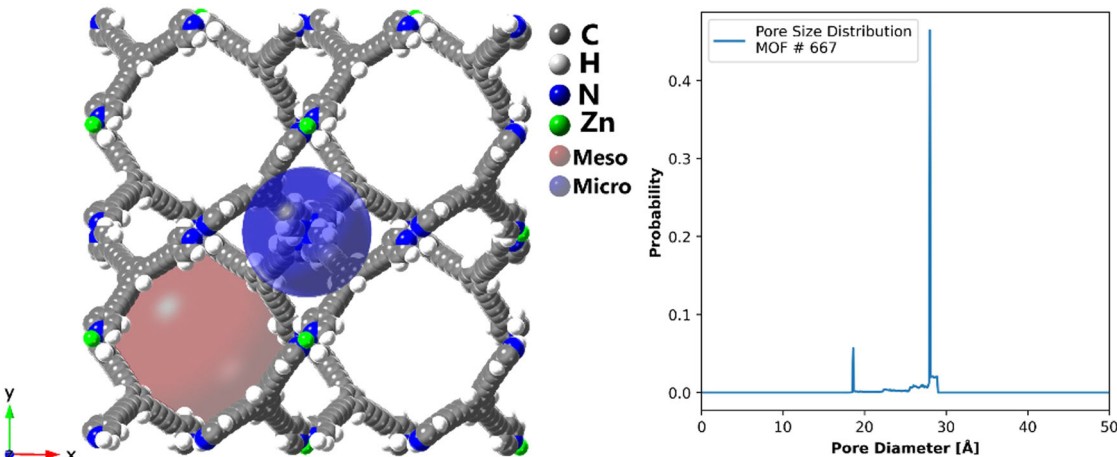

**Fig. 1 Structure of MOF #667.** Structure of MOF #667 from the ToBaCCo-1.0 database[50]. The image was generated using crystalMaker. The large transparent blue and red spheres represent the micro- and mesopores, with diameters of 18.6 Å and 27.8 Å, respectively. The pore size distribution is shown in the graph on the right. To show the micropore (blue) in one piece, we used 2 unit cells in both x and y directions (4 unit cells in total).

of pores, shown in blue and red in Fig. 1. The red mesopore has a diameter of 27.8 Å, while the blue micropore, which sits in the center of Fig. 1 connected to the surrounding mesopores, has a diameter of 18.6 Å. The structure has the bor topology, constructed from a single Zn atom metal node and a 3-connected pyridine type linker shown in Fig. S1[50]. Table S6 summarizes the textural properties of this MOF calculated via Zeo++[51].

We conducted GCMC simulations and canonical Monte Carlo simulations with Widom insertions for methane, ethane, and propane at different reduced temperatures. The reduced temperatures $T_r$ are reported based on the critical temperatures for these species from experiment, which are very close to the values predicted by the TraPPE force field as shown in Table S7. Simulations were also performed for n-hexane at 298 K ($T_r = 0.587$)[52]. The simulation input files for conducting Widom insertions in the canonical ensemble are provided as supplementary data files. The python scripts for processing the canonical isotherm data and calculating the spinodal and binodals are provided as well.

Figure 2a presents the isotherms for methane at 112 K, which corresponds to a reduced temperature of 0.587. The blue and orange points represent the results from the GCMC simulation of adsorption and desorption, respectively. The GCMC simulations predict that the adsorption and desorption isotherms have two steps, as highlighted in the ovals in Fig. 2a. Figure S2 also shows the adsorption and desorption isotherm, zoomed in between 0.2 and 0.5 bar, for methane at 112 K (same as Fig. 2a). The small step below 200 cm$^3$/cm$^3$ exhibits a very small hysteresis loop that can only be observed in Fig. S2. This small step corresponds to the filling of the micropore, as shown by the snapshot in Fig. S3 (left). The second step, which corresponds to the filling of the mesopore between 300 and 550 cm$^3$/cm$^3$, exhibits a clear hysteresis loop, and the adsorption and desorption lines do not trace each other. In addition to the GCMC isotherms, Fig. 2a shows the results from the canonical Monte Carlo simulations with Widom insertions as red points. In these calculations, we simulated the system at a series of fixed loadings from close to an empty framework to the saturation loading. At low loadings, the canonical isotherm results agree well with the GCMC results. Near the hysteresis region of the GCMC isotherm, the canonical isotherm shows an S-shaped vdW loop. From the vdW loop, we calculated the equilibrium transition fugacity (binodal), shown as a black dashed line, from the Maxwell construction of equal areas. The turning points of this vdW loop are the spinodals, which dictate the stability limits of the system: the

system is metastable between the binodal and the spinodal, while the states between the two spinodals are unstable. The adsorption step in the GCMC isotherm is located between the binodal and the low-density spinodal, while the desorption step is between the binodal and the high-density spinodal. For a schematic of the vdW loop, see Fig. S4.

At higher temperature (Fig. 2b), the vdW loop from the Widom insertions shrinks in terms of the span of loadings, consistent with the expectation that as the temperature approaches the critical temperature of the bulk adsorbate, the difference between the low-density and high-density phases is blurred. Table S8 summarizes the fugacities of the spinodals of the simulations in Fig. 2, and Fig. S5 shows the simulation results at higher reduced temperatures ($T_r = 0.806$ and $0.976$). These results further validate that as temperature increases, the vdW loops diminish and eventually disappear when the temperature is close to the critical temperature of the bulk fluid. At these higher temperatures, there is no difference between the GCMC isotherm and the canonical isotherm from NVT+Widom simulations.

Figure 2c shows the results for ethane at a reduced temperature of 0.587. The adsorption branch from GCMC is composed of two sharp step changes and a plateau region between 130 and 160 cm$^3$/cm$^3$. The snapshots that correspond to the spinodals of the two vdW loops are shown in Fig. S3 (right). The desorption isotherm from GCMC has a single, large step from ~340 cm$^3$/cm$^3$ to ~10 cm$^3$/cm$^3$. The different numbers of steps in the adsorption and desorption isotherms can be understood from the relative positions of the two high-density (liquid) spinodals along the canonical isotherm. The spinodals are the stability limits for the metastable states or branches we encounter during the GCMC simulations. Since the liquid spinodal for the mesopore around 330 cm$^3$/cm$^3$ occurs at lower fugacity (more to the left) than that of the micropore around 100 cm$^3$/cm$^3$, the high-density branch for the mesopore is more stable than that of the micropore. Thus, the desorption skips the state where the micropore is still filled and runs all the way to the almost empty MOF. Struckhoff et al.[48] reported a double-step in the isotherm with a hysteresis loop for argon in MIL-101 (Cr) at 65 K in their study. However, to the best of our knowledge, there is no literature on the type of hysteresis loop in MOFs similar to what we report having two steps for the adsorption isotherm and only one desorption step, either computationally or experimentally.

Results for propane at the four reduced temperatures (0.587, 0.701, 0.806, 0.976) are shown in Fig. S6. Comparing methane

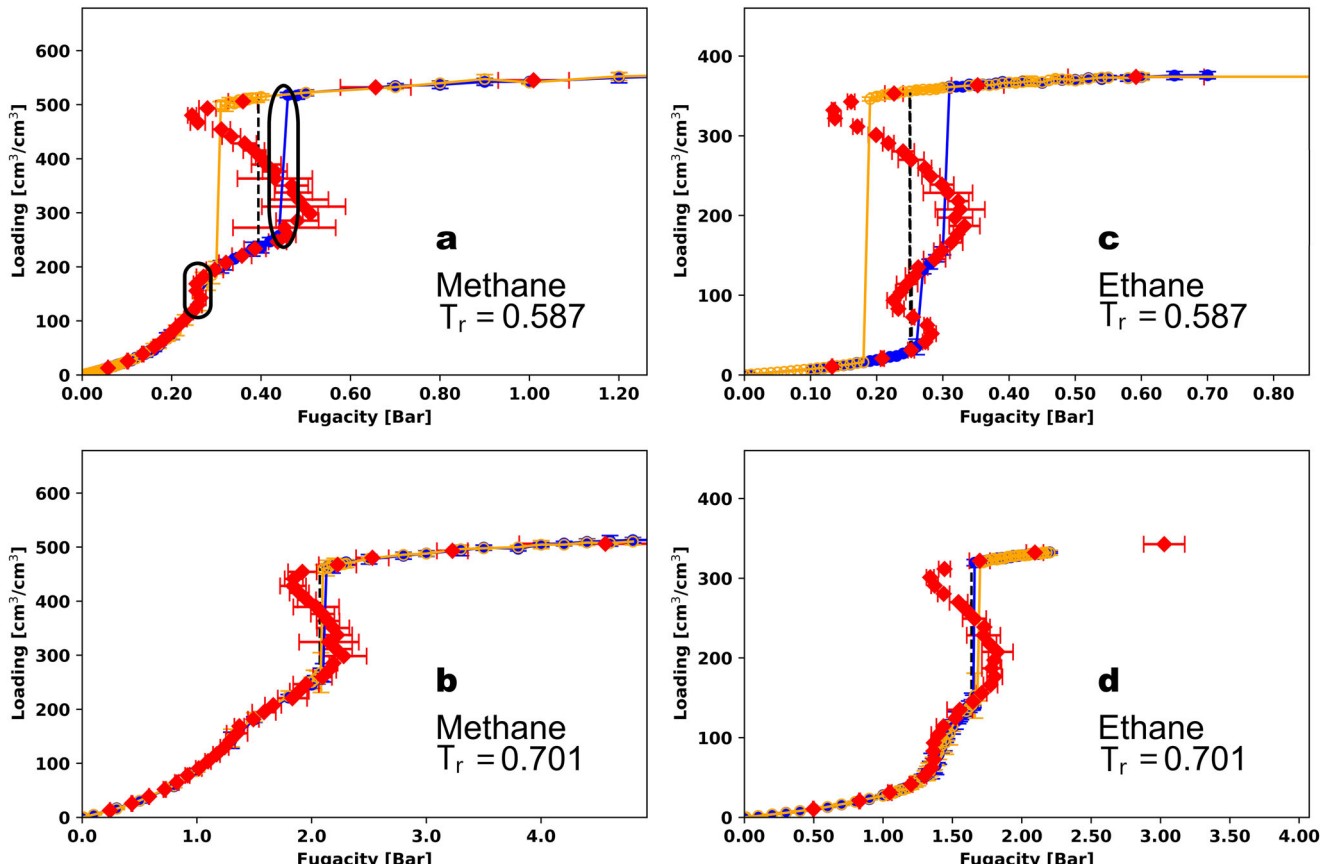

**Fig. 2 Methane and ethane isotherms in MOF #667.** Methane (**a**, **b**) and ethane (**c**, **d**) isotherms in MOF #667[50] at 112 K (**a**), 134 K (**b**), 179 K (**c**), and 214 K (**d**), corresponding to reduced temperatures of 0.587 (**a**, **c**) and 0.701 (**b**, **d**). The blue and yellow points are the adsorption and desorption isotherms, respectively, from GCMC simulations, and the red points are the canonical isotherms. The error bars show twice the standard deviation in the predictions from the GCMC and canonical simulations. Black lines are the binodal transition calculated from Maxwell's construction of equal areas. Simulations were performed using 1 unit cell of the MOF.

(Fig. 2a), ethane (Fig. 2c), and propane (Fig. S6a) at the same reduced temperature of 0.587, we can see that the vdW loops grow larger from methane to ethane to propane. Especially going from methane to ethane, we observe one big vdW loop for the mesopore and a small one for the micropore for methane at $T_r = 0.587$, while for ethane there is a big loop for each pore. Among the IUPAC classifications for hysteresis loops[53], the hysteresis loop shape for ethane in Fig. 2c and for propane in Fig. S6a is closest to the H2(a) classification. However, there are several qualitative differences in the shapes, including a mismatch between the number of steps in the adsorption and desorption branches.

Isotherms for n-hexane at 298 K ($T_r = 0.587$) from GCMC and NVT+Widom simulations are shown in Fig. 3. Similar to ethane and propane at the same reduced temperature, there are two vdW loops, but the vdW loops for the micropore and mesopore overlap. In addition, there is only one sharp step in the GCMC adsorption isotherm with no intermediate plateau, whereas methane, ethane, and propane display an intermediate plateau in their isotherms at the same reduced temperature. The reason for this difference can be understood from the relative fugacities of the spinodals. For n-hexane the low-density micropore spinodal is at a higher fugacity (more to the right in Fig. 3) than that of the mesopore, while for the other molecules, the relative fugacities are reversed.

In addition to explaining the steps in the GCMC isotherms, the NVT+Widom simulations can be used to determine the equilibria between high- and low-density phases within and between different pores in a framework. To do this, we performed the NVT+Widom calculations at different temperatures for methane and ethane and

plotted temperature versus the binodal loadings calculated from the canonical isotherm for each temperature. The results for methane and ethane in MOF #667 are shown in Fig. 4. The difference between the equilibria for methane and for ethane is mainly the size of the coexistence region for the micropore (blue dots in Fig. 4). For ethane, the high-density branch for the micropore overlaps with the low-density branch of the mesopore at low temperatures. This indicates a competition between the phases in the two pores. For methane, this equilibrium for the micropore does not overlap with the one for the mesopore. This is an additional piece of evidence for our speculation that as molecule size increases, the coexistence region for the micropore grows bigger and eventually merges with that of the mesopore. Another difference is the overall equilibrium for ethane at low temperatures (green dots in Fig. 4 right). Three binodals can be observed for the micropore, mesopore, and the overall structure. There is no overall equilibrium for methane at low temperatures. Figure S7 (right) shows the canonical isotherm for ethane at 159 K ($T_r = 0.52$). We can see three binodals for the micropore, mesopore, and the overall structure. For ethane from Fig. 4, we see that the overall equilibrium starts to appear when the low-density branch of the mesopore (red) overlaps with the high-density branch of the micropore (blue), below 192 K. When the temperature is low enough (bottom points on the right $T = 152$ K and $T_r = 0.5$), there are no red dots for that temperature (as highlighted using the orange box), meaning that there is no binodal just for the mesopore itself, and the overall binodal (green) takes over. As shown in Fig. S7 (left), at $T_r = 0.5$, there are only two binodals.

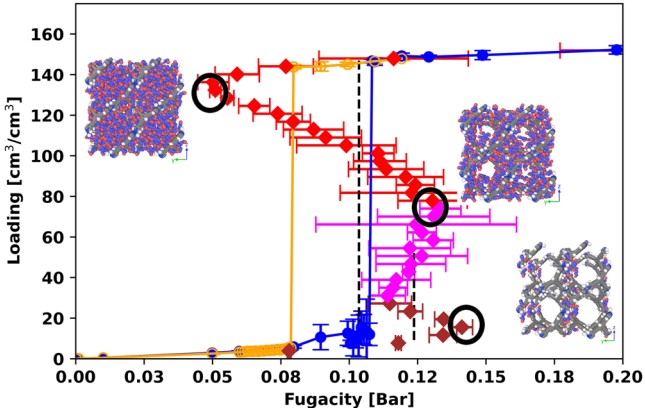

**Fig. 3 n-Hexane isotherms in MOF #667.** n-Hexane isotherms in MOF #667 at 298 K from GCMC and NVT+Widom simulations using the same color scheme as Fig. 2. The error bars show twice the standard deviation in the predictions from the GCMC and canonical simulations. Snapshots are shown at low, medium, and high loadings (isotherm points marked with black circles). Simulations were performed using 1 unit cell of the MOF, and the snapshots are duplicated into 4 unit cells for better display. In the snapshots, the pink and blue spheres are the CH₃ and CH₂ pseudo-atoms in the hexane molecules. We also colored different regions of the canonical isotherm by the pore type that it belongs to. The portion of the isotherm corresponding to filling the micropore is colored brown, the portion corresponding to filling the mesopore is colored red, and the portion where both the micropore and the mesopore are filling is colored magenta.

We also analyzed the isosteric heat of adsorption, $Q_{st}$, along the adsorption and desorption branches of the GCMC isotherms. In GCMC, $Q_{st}$ can be calculated from the fluctuations of the number of adsorbates and the system energy through[54]:

$$Q_{st} = k_B T - (\langle V*N \rangle - \langle V \rangle \langle N \rangle)/(\langle N^2 \rangle - \langle N \rangle \langle N \rangle),$$

where V is the potential energy of the system. Figure 5 summarizes the isotherms and heats of adsorption for methane and ethane in MOF # 667 at $T_r = 0.5$. The major difference in the isotherms is that for methane the adsorption has two steps, while the ethane isotherm has only one step. There are also steps in the isosteric heats at the fugacities corresponding to the step changes in the loadings, for example at a fugacity of 0.045 bar on the adsorption isotherm for methane. Interestingly, for methane, after the first jump at 0.045 bar, $Q_{st}$ decreases before the second step at a fugacity of 0.085 bar.

We then compared the host-guest and guest-guest interaction energies for methane at 95 K ($T_r = 0.5$). As shown in Fig. S8, at a fugacity of 0.068 bar, the average guest-guest and host-guest interaction energies cross. The host-guest interaction energy dominates below this fugacity, and above this fugacity the guest-guest interactions are larger in magnitude. Figure S8 also shows snapshots of the system at several points along the isotherm, which show that this crossing point is around the formation of the first layer of the mesopore. After the first layer of methane in the mesopore is formed, additional methane molecules do not interact strongly with the surface and the guest-guest interaction energy dominates the adsorption after this point.

It is known that for many systems that exhibit phase transitions, the system size affects how intense the transition is. For example, as system size increases, the order-disorder transition for the Ising model changes from a smooth transition to a sharp step change[55]. In the adsorption field, a few previous works have studied similar problems[37]. For example, Gor et al. studied the adsorption and hysteresis of a Lennard-Jones fluid in two overlapping spherical pores and found that the mechanism for

adsorption in two spherical pores differs from a system with only one spherical pore[37]. They also found that the adsorption/desorption behavior varies with the window size connecting the overlapping spheres.

To study how system size affects the GCMC and canonical isotherms, we compared the simulation results for methane in MOF #667 at 95 K ($T_r = 0.5$) in 1, 2, 4, and 8 unit cells. The results are shown in Fig. 6. We can see that as the number of unit cells increases, the single vdW loop in the mesopore for a simulation with one unit cell is broken up into 2, 4, and 8 smaller loops. Snapshots from the simulation with 4 unit cells are shown in Fig. 7. These snapshots at the 4 spinodals show that the 4 smaller loops for the mesopore correspond to the individual filling of the 4 mesopores in the simulation box. Using Widom insertions in the canonical ensemble made it possible to capture these unstable states, which are impossible to maintain in a GCMC simulation. The canonical isotherm results for the micropore region are shown in Fig. S9. It can be seen that as system size increases from one unit cell to eight unit cells, the vdW loop for the micropore gets smaller.

We also calculated the grand potential by integrating along the canonical isotherms:

$$\Omega = - \int N d\mu \qquad (2)$$

The results are shown in Fig. 8. The arrows in Fig. 8a show the direction of the grand potential with increasing loading. A purple arrow indicates that the curve is in the stable or metastable region, while a green arrow indicates it is in the unstable region (between the spinodals). As the number of unit cells increases, the number of vertices or turning points on the free energy diagram increases. These vertices represent the spinodals on the canonical isotherms. The vertices, however, span smaller ranges of the grand potential (less "stretched" in the plots) when the number of unit cells increases. This illustrates how the vdW loops depend on the system size: as system size increases, the spinodals are shifted towards the binodal transition. The intersections that are formed by two line segments that are pointing down (decreasing in the free energy, purple arrow) are the coexistence points. The overall binodal transition for the whole structure (including micro- and mesopores) is the intersection of the first and last line segments that are not in the unstable region. The overall binodal transition fugacities are shown in Fig. 6 with the blue dashed lines. As shown in Fig. 6 and Fig. 8, no matter how many boxes are used, the overall binodal transition for the whole structure is constant near ln(fugacity) ~ −2.8, which is around 0.06 bar.

The corresponding plots of grand potential versus loading are shown in Fig. S10 and provide information on the free energy barriers between the different states. We can see that as the number of unit cells increases, the free energy barriers decrease. This means that as the system size increases, it becomes easier to fill another pore. In the unstable region, which is between 260 and 500 cm³/cm³, the overall curve becomes flatter as the number of unit cells increases (comparing blue and red curves in Fig. S10). To better understand how the behavior changes as we go to a macroscopic number of unit cells, we plotted the fugacities of the lowest low-density (most bottom right red diamond in Fig. 6) and the highest high-density (most top left red diamond in Fig. 6) spinodals as a function of the number of unit cells in Fig. S11. These two spinodals are the stability limits, and they define the metastable region of the high-density and low-density phases. We can see that these two curves quickly flatten, and the fugacities converge to around 0.09 bar and 0.04 bar for the lowest low-density and the highest high-density spinodals, respectively. Thus, we anticipate that as more and more unit cells are used, the first low-density spinodal and the last high-density spinodal will

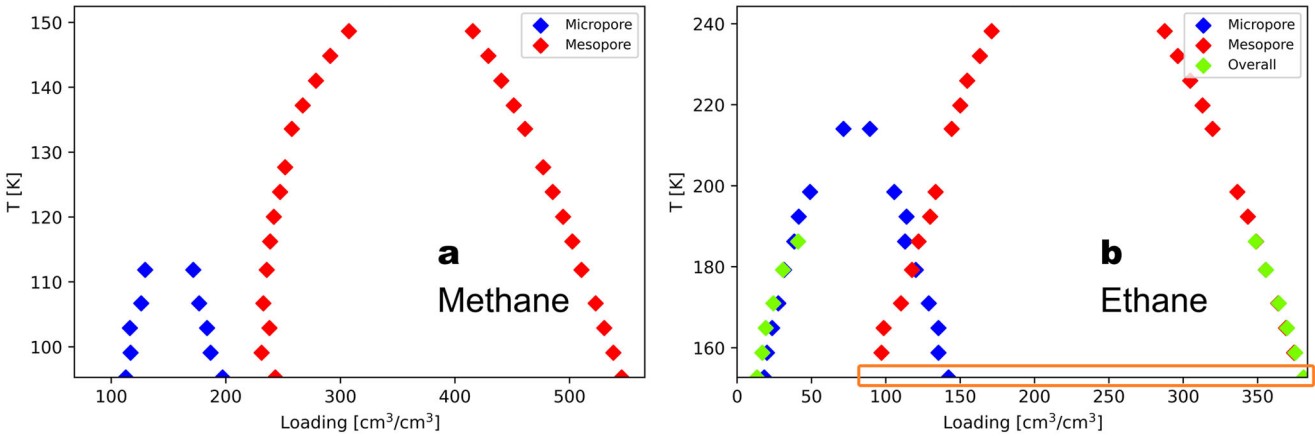

**Fig. 4 Phase equilibrium for methane and ethane in MOF #667.** Phase equilibrium for (**a**) methane and (**b**) ethane in MOF #667 from simulations using 1 unit cell. The phase equilibria for the micro- and mesopores are plotted. At the lowest temperature for ethane (153 K), there is no phase equilibrium for the mesopore, but there is an equilibrium for the micropore (blue) and the whole system (green). To highlight that there is no red dot for ethane at 152 K, we boxed the region using an orange rectangle.

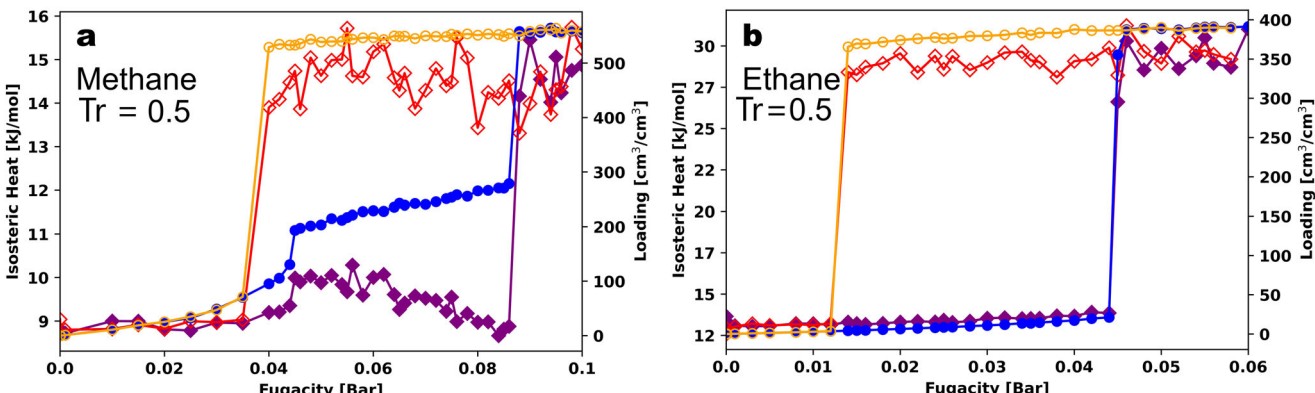

**Fig. 5 Isosteric heats for methane in MOF #667.** GCMC isotherms (right axis) and isosteric heats (left axis) for adsorption and desorption at $T_r = 0.5$ for (**a**) methane ($T = 95$ K) and (**b**) ethane ($T = 152$ K) in MOF #667. The blue and orange points are the GCMC loadings for adsorption and desorption, while the purple and red diamonds are the isosteric heats for adsorption and desorption, respectively.

remain around their converged fugacities, while the spinodals in the unstable region will converge to a straight line at the binodal. The binodal for adsorbates in a macroscopic number of unit cells, although it looks like a vertical line, is composed of many small vdW loops. Figure S12 shows a schematic of the extrapolated canonical isotherm for methane in MOF #667 at 95 K with a macroscopic number of unit cells.

The literature on molecular simulation of adsorption contains varying advice and conventions about the use of tail corrections in the energy calculation. On one hand, Smit et al. advocate the use of tail corrections because they found that using them decreases the sensitivity of simulation results to the details of truncation[56]. On the other hand, Maginn et al. argued that adsorbates in a crystalline material do not fit the picture underlying the calculation of tail corrections, which assumes that the pair distribution function $g(r) = 1$ beyond the spherical cutoff[57]. Here, we wanted to test the effect of applying tail corrections on adsorption hysteresis and vdW loops. Our default was not to apply tail corrections, but for selected examples, we applied tail corrections to the guest-guest interaction energies. This fits with the definition of the TraPPE-UA[58,59] model for n-alkanes. For both guest-guest and host-guest interactions, a 12.8 Å cutoff was used. (This is shorter than the 14.0 Å cutoff in TraPPE, but is consistent with prior work in our group for the guest-host interaction.) We first simulated methane in MOF # 667 at 95 K

($T_r = 0.5$) with and without tail corrections. The results are shown in Fig. 9.

We can see that the general shape of the vdW loop does not change, and the relative positions of the spinodals do not change. For example, the high-density spinodal of the mesopore is still lower in fugacity compared to the micropore spinodal, which indicates that the hysteresis loop shape should not change, and, indeed, the GCMC isotherms do not change their shapes. However, for the simulations including tail corrections, the fugacities of the isotherm steps and binodals (black dashed lines) are shifted to lower fugacities. The binodal fugacities for the micropore shift from 0.045 bar without tail corrections to 0.0375 bar with tail corrections, and there is a larger shift for the mesopore coexistence fugacity: 0.065 bar without tail corrections and 0.045 bar with tail corrections. For this MOF, the mesopore vdW loop is more affected by the guest-guest tail-corrections, which is reasonable since guest-guest interactions are more important in the mesopores than in the micropores.

We then simulated propane in MOF # 667 with and without tail corrections. The results in Fig. 10 and Table S9 show that the relative positions of the spinodals change in this case. The low density spinodal for the micropore (point d) is slightly shifted from 0.064 bar to 0.059 bar when going from the simulation without tail corrections to the simulation with tail corrections. However, the low-density spinodal for the mesopore (point b) is

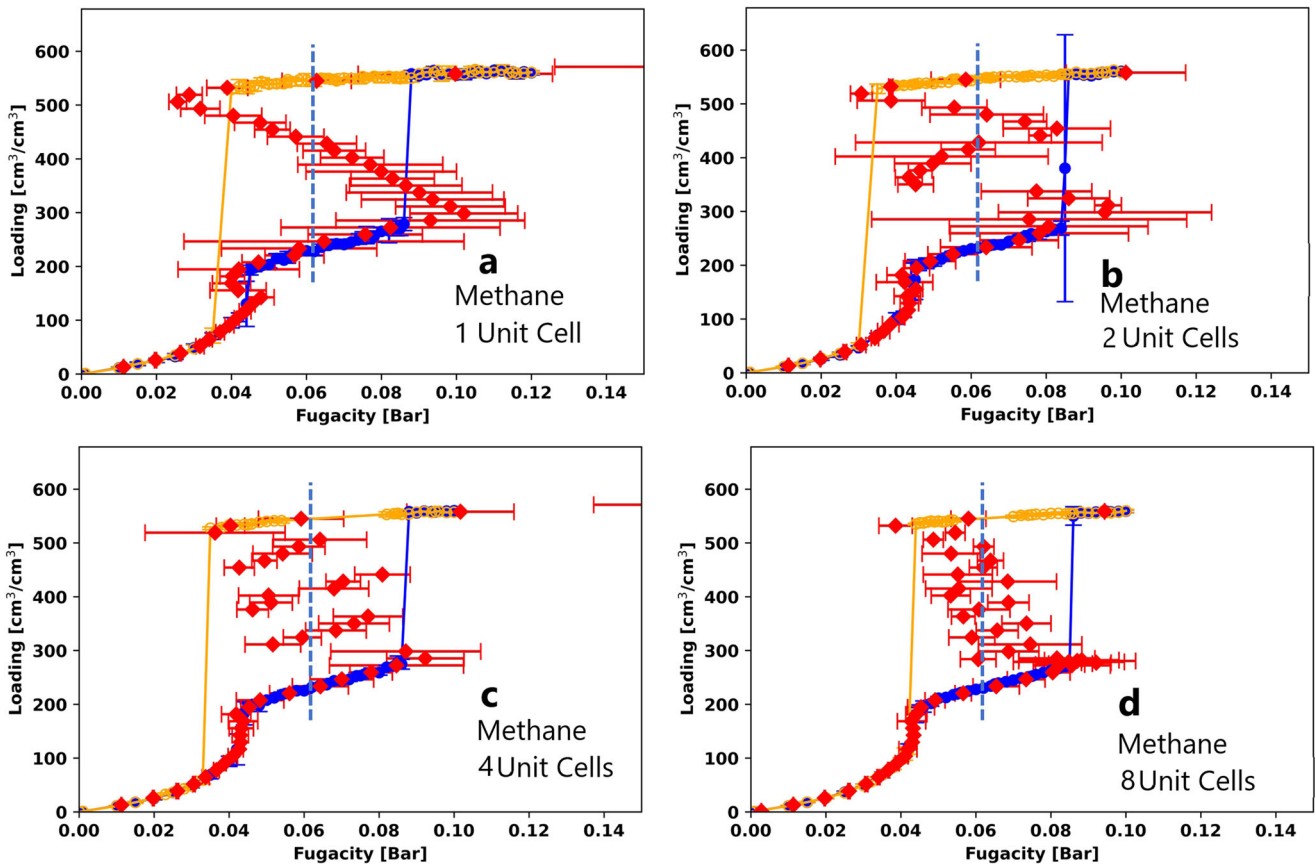

**Fig. 6 GCMC and canonical isotherms in different numbers of unit cells.** GCMC and canonical isotherms for methane in (**a**) 1, (**b**) 2, (**c**) 4, and (**d**) 8 unit cells of MOF #667 at 95 K. A zoomed-in comparison for the micropore is shown in Fig. S9. The error bars show twice the standard deviation in the predictions from the GCMC and canonical simulations.

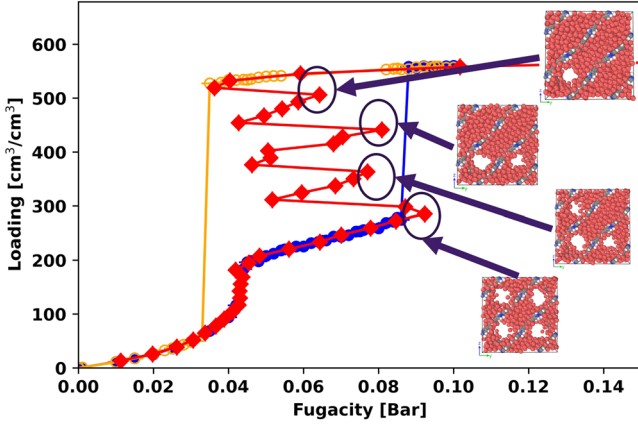

**Fig. 7 Snapshots of methane in MOF #667.** Methane isotherms in MOF #667 at 95 K ($T_r = 0.5$) from GCMC and NVT+Widom simulations using the same color scheme as Fig. 2. Snapshots are shown at selected points marked with black circles. Simulations were performed using 4 unit cells of the MOF.

strongly shifted by the tail corrections. For the middle-loading range (from 50 to 150 cm³/cm³), the S-shape almost disappears when guest–guest tail-corrections are added. However, the shape of the hysteresis loop from the GCMC simulations is unchanged. The relative positions of the spinodals determine the number of steps in the GCMC isotherms. In this case, there is one step for adsorption and one for desorption.

To better understand the shifts in the isotherms with and without tail corrections, it is instructive to normalize the fugacities

by the fugacities $f_0$ corresponding to the bulk vapor-liquid binodal, i.e., the fugacities corresponding to the vapor pressure, calculated with and without tail corrections. The isotherms with the x-axis normalized this way are shown in Fig. S13. With this normalization, the overall transition (longest vertical dashed line in left and right graphs of Fig. S13) is at the same value of $f/f_0$ around 0.21. Contrary to Fig. 10 (non-normalized fugacities), Fig. S13 shows that the low-density spinodal (point b) is not shifted but point c is shifted. For propane in MOF #667, the binodal transition fugacities and the coexistence curves are also strongly affected by the guest-guest tail-corrections as shown in Figs. S14, S15, and S16.

Finally, we briefly examined the effect of framework flexibility on adsorption hysteresis. As an example, we chose methane in IRMOF-1 due to the availability of a tested force field for describing the MOF flexibility. We applied the force field from Dubbeldam et al.[60] to describe the internal degrees of freedom of the structure, although we used a 12.8 Å cutoff other than the 12 Å cutoff from the original force field. For the rigid framework simulations, we adopted the LJ parameters from Dubbeldam et al.[61] Hybrid MC moves were included for GCMC simulations with the flexible IRMOF-1 model. The results in Figs. 11 and S17 show that flexibility reduces the size of the vdW loop and shifts the coexistence fugacity. The vdW loop for the rigid model encompasses a wider range of fugacities compared to the vdW loop for the flexible framework (Fig. 11 (left)). The adsorption isotherms from GCMC versus hybrid GCMC also display a small shift of 40 pascal in the location of the isotherm step when framework flexibility is included.

From Fig. S17, we can see that the vdW loops using both the rigid and the flexible framework models occur at loadings below

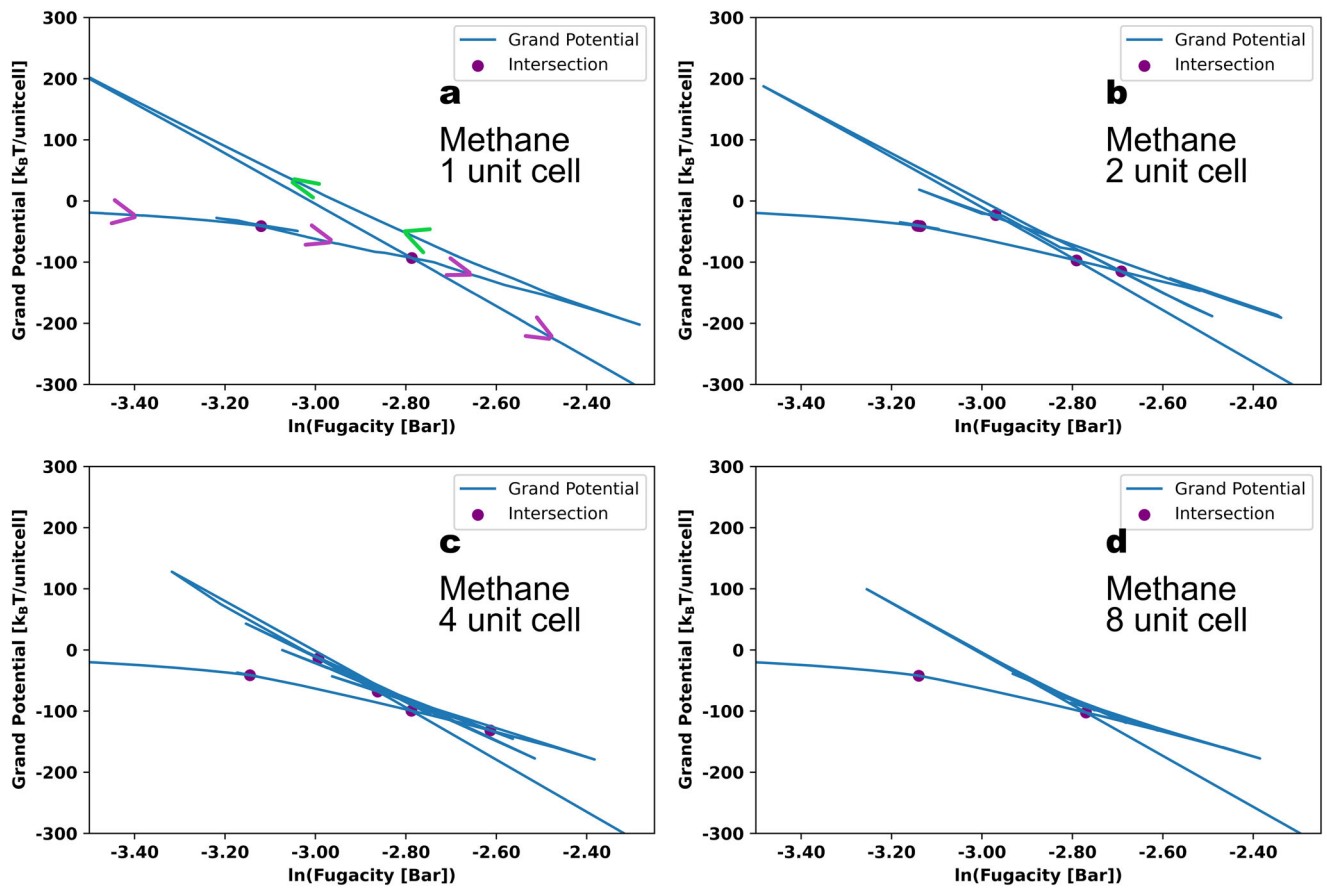

**Fig. 8 Grand potentials of methane in MOF #667.** Grand potential versus natural log of the fugacity for methane in MOF # 667 at 95 K for (**a**) 1, (**b**) 2, (**c**) 4, and (**d**) 8 unit cells. The purple dots indicate the binodal formed by two line-segments on the free energy plot with the same direction. For plot (**a**), we added arrows in the direction of increasing number of molecules, with purple arrows for the stable and metastable regions and green arrows for the unstable regions. For (**d**), we only show the binodal fugacities for the micropore and the overall equilibrium for the mesopores.

the pore-filling in the typical capillary condensation scheme. The vdW loop is between 43 cm$^3$/cm$^3$ (20 molec/uc) and 259 cm$^3$/cm$^3$ (120 molec/uc), while the saturation loading for IRMOF-1 is around 453 cm$^3$/cm$^3$ (210 molecules per unit cell). Figure S18 shows some snapshots from the MD + Widom simulations for methane in rigid frameworks of IRMOF-1 at 112 K at 108 cm$^3$/cm$^3$ (50 molec/uc), 237 cm$^3$/cm$^3$ (110 molec/uc), and 432 cm$^3$/cm$^3$ (200 molec/uc). Loadings of 108 cm$^3$/cm$^3$ (50 molec/uc) and 237 cm$^3$/cm$^3$ (110 molec/uc) correspond to the low-density and high-density spinodal of the vdW loop respectively, while 432 cm$^3$/cm$^3$ (200 molec/uc) is close to the saturation loading of IRMOF-1. The snapshots show that at the low-density spinodal, the methane molecules adsorb near the nodes of the MOF; at the high-density spinodal, the methane molecules cover the first layer of the MOF; and at 432 cm$^3$/cm$^3$ (200 molec/uc), the MOF is saturated with methane molecules and there is almost no void space. We compared the transition we found with the work of Höft et al.[22] In their work, they found a surface transition, i.e., a transition at low densities that is associated with the coverage of the first layer, for methane in IRMOF-1 below 114 K, and the filling of the pore of IRMOF-1 happens below 110 K. They also found that the surface transition is between 43 cm$^3$/cm$^3$ (20 molec/uc) and 285 cm$^3$/cm$^3$ (132 molec/uc), and the pore-filling is from 324 cm$^3$/cm$^3$ (150 molec/uc) to 454 cm$^3$/cm$^3$ (210 molec/uc). We see that the vdW loop we simulated falls into the range of the surface transition, and the smooth pore-filling is between 324 cm$^3$/cm$^3$ (150 molec/uc) and 454 cm$^3$/cm$^3$ (210 molec/uc). What we

observed agrees with the observation from Höft et al.[22] Thus, we classify the vdW loop to be the surface transition.

To investigate the underlying reason for the shift in the isotherms when including framework flexibility, we analyzed the structures of IRMOF-1 from the canonical isotherm simulations at 237 cm$^3$/cm$^3$ (110 molec/uc) using both the rigid and flexible framework representations at the end of the NVT simulations. Snapshots comparing the rigid and flexible framework of IRMOF-1 in Fig. S19 show that there are slight differences between the two representations. While the positions of the zinc atoms remained essentially unchanged, there are changes in the orientations of the linkers, including the oxygen atoms of the BDC linkers. We then analyzed the host-guest and guest-guest interactions calculated by the canonical ensemble molecular dynamics simulations at different loadings of methane using the rigid and flexible framework representations. The results in Fig. S20 show that for the guest-guest interaction, the difference is negligible. However, the host-guest interaction energy of the flexible model is always higher (less favorable) than the rigid model, which is consistent with the isotherms being shifted to the right when framework flexibility is included. We also added the detailed analysis of the difference in host-guest energy per molecule in Fig. S21.

By investigating the host-guest energy difference and comparing the canonical isotherms, we see that the flexibility of the framework is coupled with the phase transitions of adsorbates in IRMOF-1. The first transition has a vdW loop, and framework flexibility makes the vdW loop smaller. In addition, flexibility makes the host-guest interaction less favorable and shifts the vdW loop to higher

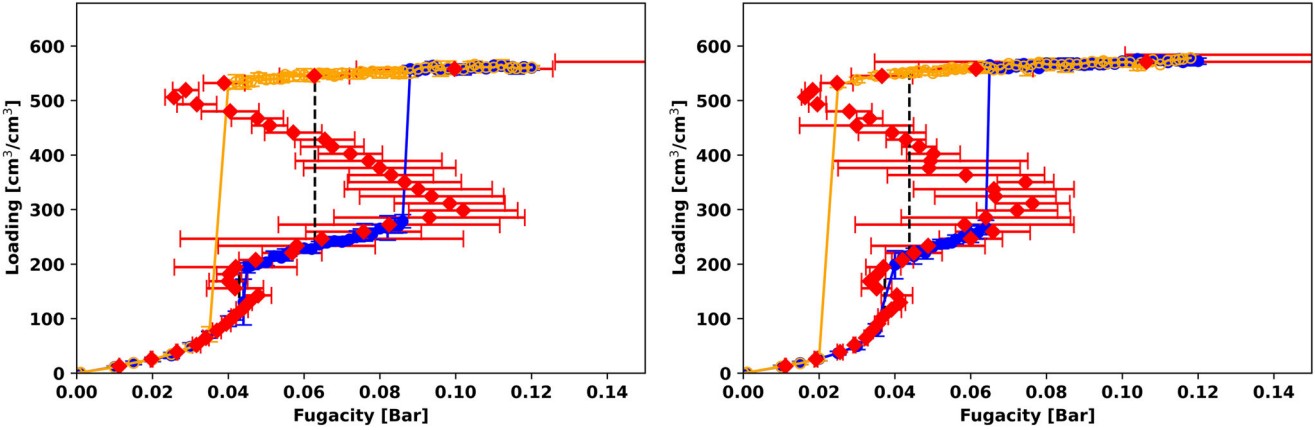

**Fig. 9 Methane isotherms with and without tail-corrections.** Methane isotherms in MOF #667 using 1 unit cell at 95 K ($T_r = 0.5$) without (left) and with (right) tail corrections for the guest–guest interaction. The error bars show twice the standard deviation in the predictions from the GCMC and canonical simulations.

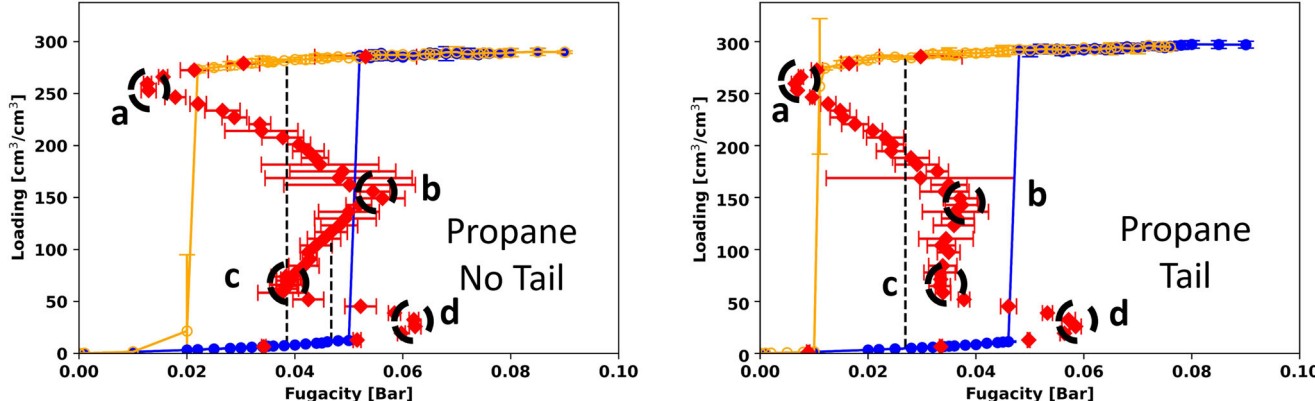

**Fig. 10 Propane isotherms in MOF #667 with and without tail correction.** Propane isotherms in MOF #667 using 1 unit cell at 185 K ($T_r = 0.5$) without (left) and with (right) tail corrections for the guest-guest interactions. The error bars show twice the standard deviation in the predictions from the GCMC and canonical simulations. The vertical dashed lines show the binodal transitions for the micropore and the overall transition for the whole range of loadings. There is no mesopore equilibrium transition for this temperature.

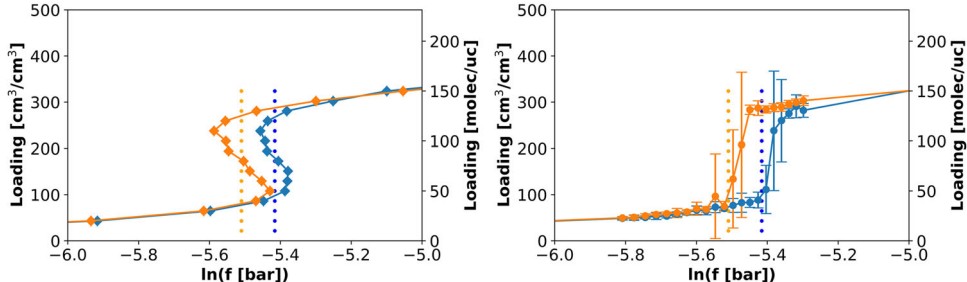

**Fig. 11 Methane isotherm in IRMOF-1.** MD + Widom canonical isotherms (left) and GCMC/hGCMC adsorption isotherms (right) for methane at 112 K ($T_r = 0.587$) in both rigid (orange) and flexible (blue) structure models of IRMOF-1. The error bars in the plot on the right-hand side show twice the standard deviation in the predictions from the GCMC. The dotted lines are the corresponding binodal transitions for both framework models. The graphs are zoomed into the fugacity range of the vdW loop, and the full isotherms are shown in Fig. S17.

fugacities. Since the first transition is related to methane forming the first layer on the surface of the framework, we suspect that the metal flexibility of the framework disrupts the surface integrity and makes it more difficult for methane to form a surface layer. For the second transition, the system reaches saturation through a smooth transition, and the framework flexibility causes steric hindrance and leads to a smaller saturation loading. From this example, we can see that framework flexibility can influence and couple with the phase behavior of the adsorbate.

## Conclusions

In this work, we examined the adsorption of n-alkanes in a metal-organic framework that has interconnected micropores and meso-pores and exhibits complex phase behavior. In addition to standard grand canonical Monte Carlo simulations, we used Widom insertions in the canonical ensemble to predict the adsorption of methane, ethane, propane, and n-hexane. At low temperatures, GCMC predicts sharp steps in the isotherms accompanied by adsorption hysteresis. Examining the molecular-level siting revealed

that the steps and hysteresis are associated with filling of the micropores and mesopores. The Widom insertions allowed us to trace out a van der Waals loop associated with each hysteresis loop. We discovered that, as molecule size increases or temperature decreases, the vdW loops grow bigger. We also studied how the coexistence curves and binodals of different pores in the MOF interact at various temperatures.

The simulations revealed interesting system size effects. If the simulation contains a single mesopore, the isotherms have a single vdW loop associated with filling the mesopore. If more unit cells are used, additional vdW loops appear, each associated with the filling of a single mesopore. By calculating the grand potential, we determined all of the binodal transitions along the canonical isotherms and found that the overall binodal transition stays the same as system size increases, but the vdW loop and spinodal positions change. For a macroscopic sample composed of many unit cells, the results suggest that the canonical isotherm will be the combination of many small vdW loops—one for each individual pore. The vdW loops in the unstable region will converge to the binodal, while the low-density spinodal with the lowest loading and the high-density spinodal with the highest loading will converge to certain fugacities.

We also checked the effect of guest-guest tail corrections on the adsorption hysteresis and vdW loops and found that tail corrections can have a strong effect on the vdW loops, the positions of the spinodals and binodals, and the phase-coexistence curves.

Finally, we studied the low-temperature adsorption of methane in IRMOF-1, comparing simulations with rigid and flexible framework models. At 112 K, there is a vdW loop corresponding to a surface transition where the methane molecules form the first layer on the pore walls. When incorporating framework flexibility, we found that the flexibility of IRMOF-1 disrupts the surface of the MOF and makes the host-guest interaction energy less favorable. Thus, the vdW loop for the surface transition becomes smaller and the pore condensation transition is shifted to higher fugacity. The fundamental insights from this work may help improve adsorbents for applications such as adsorption cooling, atmospheric water harvesting, and carbon capture where it is often desirable to find conditions or materials that avoid hysteresis.

## Methods

**Model**. Non-bonded interactions were modeled using a Lennard-Jones (LJ) potential between every pair of non-bonded atoms or pseudo-atoms. The Lorentz-Berthelot mixing rules were used to obtain the Lennard-Jones parameters between different atom types. All LJ parameters are summarized in Table S1. The LJ parameters for the framework atoms were taken from the Universal Force Field (UFF)[62], and the MOF atoms were held fixed during the simulations. For methane, propane, and n-hexane, we used the TraPPE United Atom (UA) force field[58], and for ethane we used TraPPE-UA[59], which provides improved predictions for ethane compared to the original TraPPE-UA model. The use of the UFF + TraPPE pair has been shown to provide good predictions of alkane adsorption in MOFs in previous work[63–65]. Unless otherwise indicated, a cutoff of 12.8 Å was used for the Lennard-Jones interactions with no tail corrections. However, for comparison, we also conducted some simulations using tail corrections for the guest-guest interactions.

**Grand canonical and canonical Monte Carlo simulations**. All molecular simulations were performed using RASPA-2.0[66]. For GCMC simulations of methane, we used 10,000 initialization and 10,000 production cycles. In RASPA-2.0[66], a cycle is composed of $N_A$ steps, where $N_A$ is equal to the maximum of 20 and the number of adsorbate molecules in the simulation box. Translation, re-insertion, and swap moves were chosen randomly with equal probability. In RASPA, a swap move randomly attempts an insertion or a deletion move with equal probability. For ethane, propane, and n-hexane, 30,000 initialization and 30,000 production cycles were used. In addition to the moves used for methane, rotation and cut-and-regrow moves were used. For n-hexane, a configurational-bias/continuous-fractional component (CB/CFC)[67] move was added. The GCMC adsorption and desorption isotherms were calculated sequentially "up and down" the isotherm by starting each simulation using the final configuration from the previous pressure point.

For the canonical isotherms, NVT+Widom simulations were used with at least 10,000 initialization cycles and 1,000,000 production cycles. The simulations used the same moves as for GCMC for methane, ethane, propane, and hexane except

that swap (insertion/deletion) and CB/CFC moves were not used. Widom insertion moves (described below) were added to the canonical simulations with twice the probability of the other moves.

We also performed simulations for methane in IRMOF-1 using a previously published model that incorporates framework flexibility[60]. We shifted the vdW interactions between the pseudo atoms to follow the definition by Dubbeldam et al.[60] A detailed summary of the force field parameters including the vdW and bonding terms is provided in Tables S1–S5. Here, we used canonical molecular dynamics (MD) simulations with a Nose-Hoover thermostat[68] instead of canonical Monte Carlo when calculating the canonical isotherm as a more convenient way to sample the flexibility of the framework. We used 1.5 million steps, with a time step of 2 femtoseconds. In RASPA[66], the MD simulations used a smoothing function for the truncated potentials. For the canonical MD simulations, we used 200 Widom insertions at every step. For the GCMC simulations with a flexible framework, we added a hybrid MD move to the list of moves, with all moves chosen randomly with equal probability. For the hybrid MD move, we used 5 molecular dynamics steps with a time step of 0.5 femtosecond.

**Widom insertions**. In the original Widom insertion scheme[69], the excess chemical potential ($\mu_{ex}$) at temperature $T$ is calculated as

$$\mu_{ex} = -k_B T \ln \langle \exp(V_{test}/k_B T) \rangle \tag{3}$$

where $k_B$ represents Boltzmann's constant, $V_{test}$ is the potential energy felt by the inserted test particle, and the angular brackets indicate an ensemble average. When combined with CBMC, $\mu_{ex}$ is given by[70]

$$\mu_{ex} = -k_B T \ln \langle W_r \rangle \tag{4}$$

where $W_r$ is the average Rosenbluth factor, which is given by

$$W_r = \prod_i^s w(i) \tag{5}$$

where $w(i)$ is the Rosenbluth weight of the $i^{th}$ segment of the probe molecule, given that there are s segments in the molecule. For the growth of each segment, 10 trial positions were randomly generated, so that $w(i) = \sum_{l=1}^{10} \exp(-V_{i,l}/k_B T)$. Here the sum is over the Boltzmann factor of the potential energy for the $l^{th}$ trial position of the $i^{th}$ segment. More information on the implementation of CBMC in the RASPA code is given by Dubbeldam et al.[71] (Note that configuration bias is used for most moves in RASPA, not just Widom insertions.)

After the simulation is finished, the calculated chemical potential is converted to fugacity $f$ by[72]

$$f = \frac{W_{IG} k_B T N}{V} \exp(\mu_{ex}/k_B T), \tag{6}$$

where $N$ is the number of molecules in the system, $V$ is the volume of the system, and $W_{IG}$ is the ideal gas Rosenbluth weight of the molecule. Substituting $\mu_{ex}$ from Eq. 4 yields

$$f = \frac{W_{IG} k_B T N}{V \langle W_r \rangle}. \tag{7}$$

## Data availability

The simulation data and python scripts are available at https://figshare.com/articles/dataset/Data/21743354/2. Sample input files for RASPA are included as supplementary data files.

## Code availability

The python scripts for data processing and calculating the binodal from the van der Waals loop are available at https://figshare.com/articles/dataset/Data/21743354/2.

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

## Acknowledgements

This work was supported by the U.S. Department of Energy, Office of Basic Energy Sciences, Division of Chemical Sciences, Geosciences and Biosciences under Award No. DE-SC0023454. Z.L. acknowledges support from a Data Science Fellowship via the Northwestern Institute on Complex Systems (NICO). This research was partly supported through the computational resources provided for the Quest high-performance computing facility at Northwestern University, which is jointly supported by the Office of the Provost, the Office for Research, and Northwestern University Information Technology. This research used resources of the National Energy Research Scientific Computing Center, a DOE Office of Science User Facility supported by the Office of Science of the U.S. Department of Energy under Contract No. DE-AC02-05CH11231 using NERSC award BES-ERCAP0023154. Z.L. thanks Kunhuan Liu for helpful discussions. The authors also thank the anonymous reviewers for their helpful suggestions.

## Author contributions

Z.L. and R.Q.S. designed the concept of the paper. Z.L. and J.T. performed the simulations. Z.L. wrote the paper. R.Q.S. supervised the project and revised the paper.

## Competing interests

R.Q.S. has a financial interest in the startup company NuMat Technologies, which is seeking to commercialize metal–organic frameworks. Z.L. and J.T. declare no competing interests.
