## [Peer Review File · Communications Chemistry]

Computational investigation of hysteresis and phase equilibria of n-alkanes in a metal-organic framework with both micropores and mesoporesReviewers' comments:

Reviewer #1 (Remarks to the Author):

The paper by Li et al. presents the results of a study on hysteresis and phase transitions of alkanes in MOFs, using the Widom particle insertion method together with canonical Monte Carlo simulations to trace the van der Waals loops for phase coexistence within the pore space of the MOF. The work is rigorous and technically sound. However, in my opinion, the novelty of the work is not clearly articulated in comparison with prior publications on this topic (see below for details). As such, I recommend a major review before the paper is considered for publication.

1. The introduction of the paper should be expanded to discuss prior applications of similar simulation methods (most notably the gauge-cell method of Neimark and Vishnyakov, but also other approaches like umbrella sampling) to elucidate the phase equilibrium, phase diagrams and full isotherms (including van der Waals loops) of methane in porous materials. Among others, the following references are relevant in this context for phase equilibrium of hydrocarbons in: a) model pores (10.1007/s10450-007-9006-8; 10.1080/00268970210166255; 10.1016/j.jngse.2022.104691; 10.1016/j.fuel.2021.120909; 10.1021/la0608720), heterogeneous or complex pores (10.1016/j.fluid.2017.08.017) and even Metal Organic Frameworks (10.1021/la903643f; 10.1021/jacs.5b04077). In particular, given the previous application of similar methods to MOFs, it is crucial that the authors explain what is new about the current study, and what new physical insights it brings to understand the problem of hydrocarbon confinement in porous materials.

2. In light of the above point, the authors' statement in page 3 that "previous studies [...] neglect features of real materials such as interconnections between pores and pores of different size" is not strictly correct. Apart from the papers mentioned in the previous point, this neglects a large body of work by Monson and co-workers on disordered porous materials like Vycor glass (see e.g. 10.1103/PhysRevE.67.041207 and associated references). It is important that the authors do justice to prior work in the field, and describe the state-of-the-art regarding current understanding of phase equilibrium in complex porous materials.

3. Also in page 3, the authors state that Schoen et al (ref 22) argued that hysteresis in GCMC simulations arises from a failure of the algorithm. I may be wrong, but I believe those authors' interpretation is that hysteresis arises from an inability of GCMC to adequately sample the necessary fluctuations – hence why the equilibrium curve cannot be traced. However, that is not necessarily a failure of the algorithm, since it actually mirrors (at least to some extent) what happens in experimental phase equilibrium. In any case, I think the authors' statement above is at least open to question, so should be rephrased.

4. In the hybrid MD moves used to study framework flexibility, it is unclear if the authors used a flexible or rigid model for the adsorbate molecules. If flexible, then the appropriate parameters should be reported. Also, the effect of adsorbate flexibility on the adsorption isotherms should be tested – if the latter is significant, then this needs to be decoupled from the effect of framework flexibility.

5. In the GCMC description (page 4), I believe the authors forgot to mention deletion trials – "Translation, re-insertion and swap [...] moves were chosen randomly...". Furthermore, when explaining the moves used in NVT simulations (page 5), the authors also forgot to mention insertion and deletion trials – "...except that swap and CB/CFC moves were not used."

6. A reference is missing for the Nosé-Hoover thermostat in page 5.

7. In Figure 1, the authors report a pore size distribution for MOF #667. They should explain how this was obtained, and which software was used for that purpose. Furthermore, it would be useful if the authors reported a set of common physical characteristics of the MOF, such as geometric surface area,

pore volume, limiting pore diameter, etc.

8. There is no validation of the molecular model used in the simulations. First of all, the authors should demonstrate that the models are able to predict adsorption in the chosen MOF, or (if experimental isotherms for that material are not available) at least in similar frameworks, either by carrying out additional simulations or by citing appropriate literature work. Second, and perhaps most importantly, the authors should discuss if the hysteresis behaviour observed in the simulations (namely a double-step in the isotherm) has been observed experimentally. Without this kind of validation, one cannot be certain that the observed behaviour is not an artefact of the model.

9. The study on finite-size effects (pages 14-17) is quite interesting and begs the question – what will happen in a realistic macroscopic sample of the MOF material. The authors offer a rather vague speculation in page 17, but I think this should be explored in more detail. From the data in Figure S9, it seems that the vdW loops near the centre of the hysteresis loop become smaller and closer to the equilibrium transition step. One may therefore argue that in a macroscopic material, those loops will become infinitesimally small so as to follow the real equilibrium curve (dashed blue line). However, the two (or perhaps more) vdW loops closest to the spinodal will remain as wider loops to connect to both metastable branches of the isotherm. This would be quite a distinct picture from the isotherm shown in Figure S9a for a single unit cell. Are the authors able to provide further information on the expected shape of the vdW loop as the system size tends to infinity? Is it possible to carry out an even larger simulation than 8 unit cells, or is that already at the edge of what is computationally feasible?

10. On a related point, it is not clear what happens to the vdW loop for the micropore as the system size increases. At the naked eye (Figure S9), it seems to disappear for the larger systems, but perhaps that is just due to the small scale of the figure. Can the authors examine the evolution of that loop in more detail, and again offer some interpretation of what should be expected in a realistic macroscopic sample?

11. Perhaps the most interesting part of the paper, in my opinion, is the analysis of the effect of framework flexibility (Figure 10). However, the discussion of that section is underdeveloped. For example, the authors merely state that both the NVT and GCMC isotherms are shifted to the right by ~ 40 Pa, but do not offer a possible explanation for this shift. Perhaps by analysing the movement of the framework during adsorption in more detail, some additional insight can be gained? Also, it seems that the onset of the adsorption step in GCMC coincides with the equilibrium step determined from NVT. Is this a coincidence, or is there a physical reason why this should be the case?

Reviewer #2 (Remarks to the Author):

See separate file

Reviewer #3 (Remarks to the Author):

This paper applies GCMC/MD simulations in combination with the Widom insertion method to reveal the states of the adsorbed fluid inside the adsorption hysteresis loops. The particular novelty of the paper is the application of this analysis to the mesoporous metal organic frameworks (MOFs) and various hydrocarbons. To the best of my knowledge this is the first attempt to explore the metastable states and confined phase diagrams in MOFs.

MOFs is a relatively new class of materials and we are still in the process of understanding their fundamental adsorption characteristics. The current article seems to support the notion that mesoporous materials share fundamental physics of adsorption phenomena. On the other hand, MOFs

seem to feature some new effects. For example, the increase of the number of loops in the system with the system size is quite interesting. From this perspective, I believe this paper will be of interest to a broad community of scientists working in the field of adsorption and adsorption in MOFs specifically.

The paper is clearly written, the methodology is well explained, and the references are appropriate. My recommendation is to publish this paper.

Few issues, where I think the authors could provide further comment are as follows:

1) Extrapolating the system size effects to large system, what would be the picture in this case? Do the authors expect the space inside the hysteresis loop filled in with infinite number of metastable states?

2) Will the data and the simulation setups shared using some github depository system?

Referee report for the manuscript “Computational Investigation of hysteresis and phase equilibria of n-alkanes in a metal-organic framework with both micropores and mesopores

By Snurr and co-workers.

The study presented by Snurr and co-workers deals about a very interesting subject on the adsorption behavior within nanoporous frameworks having both micro- and mesopores. For this purpose, a complementary set of simulations are performed. I believe the results obtained by the authors are important and timely for the community working on adsorption characteristics of nanoporous frameworks and in general advise to publish the paper, albeit after taking into consideration a few suggestions.

I have a few questions and suggestions for improving the paper. Various suggestions intend to clarify the main message of the paper, making it more accessible for a broader audience and emphasizing more the new findings and physical chemistry insights.

The authors use several times (already in abstract) the nomenclature “in a metal-organic framework having both micropores and mesopores”. I think it would be better to introduce the rationale for the material more convincingly. If I understood correctly, the particular material is not crucial, it only needs to have clear micro- and mesopores and a few other characteristics which are mentioned by the authors. For this purpose, the authors took a material from their ToBaCCo database. It should be clearly emphasized that one particular material has been chosen with the proper motivation, but that the material as such is not the most important aspect. This is a fundamental study to obtain more insight into the adsorption properties of micro-/mesoporous materials.

I suggest to also make the main hypothesis/research result more clear in the abstract. Now it is mentioned “At low temperatures, the calculated isotherms exhibit sharp steps accompanied by hysteresis”. If I understood correctly, they found that both at low temperatures and for increasing molecule size the van der Waals loops grow. I thus suggest to also mention the influence of the molecule size in the abstract. In general the abstract would benefit from some rewriting to make the main conclusions and research hypothesis more clear.

The introduction is well written and the references are appropriate. The only remark on page 3 is the introduction of the MOF, see my remark above.

Methods :

The authors used a series of force fields to perform the simulations. Can they comment on the dependence of the results on the particular choice of force field. Probably, for this study where it is the intention to give qualitative insights into the adsorption isotherms for hierarchical materials, the specific force field is no problem? Can the authors confirm or comment on this? It is well known that in general for the calculation of isotherms, the specific force field may be very important.

Results :

Overall, the results section is written in a technically very detailed way and the simulations are performed with care, however I invite the authors to emphasize their main physical chemistry insights and conclusions, in relation to the starting hypothesis more clearly throughout the paper.

Figure 3 could maybe be improved, it is mentioned that the vdW loops overlap for the micro- and mesopores, but it would be interesting to indicate this more clearly on the figure.

The supporting information is essential for following the paper. It might be considered to take up an additional figure in the main manuscript on the size effects, as I believe these are important conclusions. It could be Figure S9 possibly in adapted form to make it suited for the main manuscript.

The sentence "The corresponding plots of grand potential versus loading are shown in Figure S10 and provide information on the free energy barriers between the different states" on page 16, is not fully clear. Can the authors comment more on the free energy barriers?

Conclusions :

"In this work, we examined the adsorption of n-alkanes in a metal-organic framework, reference 41". It is not fully clear why reference 41 is given at this place.

The conclusions found for the system size are very interesting, it is suggested that the canonical isotherms are the combination of many small vdW loops. Could the authors comment on the possible experimental validation of these conclusions? In how far are these conclusions realistic or can these only be observed in simulations?

In the sentence "the results suggest that the canonical isotherm will be the combination of many small vdW loops – one for each pore". Please specify the type of pore.

Finally it might be interesting to give some perspective on the potential application importance of their findings. Nowadays many hierarchical materials are studied for particular applications and thus such perspective could increase the impact of the work.

Response to Reviewers' Comments:

Reviewer #1:

The paper by Li et al. presents the results of a study on hysteresis and phase transitions of alkanes in MOFs, using the Widom particle insertion method together with canonical Monte Carlo simulations to trace the van der Waals loops for phase coexistence within the pore space of the MOF. The work is rigorous and technically sound. However, in my opinion, the novelty of the work is not clearly articulated in comparison with prior publications on this topic (see below for details). As such, I recommend a major review before the paper is considered for publication.

1. The introduction of the paper should be expanded to discuss prior applications of similar simulation methods (most notably the gauge-cell method of Neimark and Vishnyakov, but also other approaches like umbrella sampling) to elucidate the phase equilibrium, phase diagrams and full isotherms (including van der Waals loops) of methane in porous materials. Among others, the following references are relevant in this context for phase equilibrium of hydrocarbons in: a) model pores (10.1007/s10450-007-9006-8; 10.1080/00268970210166255; 10.1016/j.jngse.2022.104691; 10.1016/j.fuel.2021.120909; 10.1021/la0608720), heterogeneous or complex pores (10.1016/j.fluid.2017.08.017) and even Metal Organic Frameworks (10.1021/la903643f; 10.1021/jacs.5b04077). In particular, given the previous application of similar methods to MOFs, it is crucial that the authors explain what is new about the current study, and what new physical insights it brings to understand the problem of hydrocarbon confinement in porous materials.

Answer: Thank you for these additional helpful references. We added these references to the Introduction, briefly discussed them, and expanded our previous short discussion of previous work using the gauge-cell method. The reference 10.1021/jacs.5b04077 was especially helpful, and we included several paragraphs of discussions for the flexibility of IRMOF-1 section from page 23 to page 26 in the manuscript in the light of their results.

We have modified the last paragraph of the Introduction to highlight what is new in this work and what new physical insights it brings.

2. In light of the above point, the authors' statement in page 3 that "previous studies [...] neglect features of real materials such as interconnections between pores and pores of different size" is not strictly correct. Apart from the papers mentioned in the previous point, this neglects a large body of work by Monson and co-workers on disordered porous materials like Vycor glass (see e.g. 10.1103/PhysRevE.67.041207 and associated references). It is important that the authors do justice to prior work in the field, and describe the state-of-the-art regarding current understanding of phase equilibrium in complex porous materials.

Answer: We have added references by Monson et al. and rewrote this part of the introduction. Additional lines were also added at the end of the introduction to briefly discuss the impact of this work.

3. Also in page 3, the authors state that Schoen et al (ref 22) argued that hysteresis in GCMC simulations arises from a failure of the algorithm. I may be wrong, but I believe those authors' interpretation is that hysteresis arises from an inability of GCMC to adequately sample the

necessary fluctuations – hence why the equilibrium curve cannot be traced. However, that is not necessarily a failure of the algorithm, since it actually mirrors (at least to some extent) what happens in experimental phase equilibrium. In any case, I think the authors' statement above is at least open to question, so should be rephrased.

Answer: One could argue that inadequately sampling the necessary fluctuations is a failure of the algorithm, but the reviewer makes a good point that calling it a failure is probably not sufficiently informative. We have, therefore, rewritten this sentence to match better what Schoen et al. expressed in their conclusion.

4. In the hybrid MD moves used to study framework flexibility, it is unclear if the authors used a flexible or rigid model for the adsorbate molecules. If flexible, then the appropriate parameters should be reported. Also, the effect of adsorbate flexibility on the adsorption isotherms should be tested – if the latter is significant, then this needs to be decoupled from the effect of framework flexibility.

Answer: The force field parameters for methane in IRMOF-1 with the bonded terms for capturing the flexibility of IRMOF-1 have been added to the SI. For methane adsorption in IRMOF-1, we used a united representation for methane where the methane molecule is represented as a single Lennard-Jones sphere. Thus, there is no internal flexibility within the adsorbate molecule itself, and in our study only the framework is flexible.

5. In the GCMC description (page 4), I believe the authors forgot to mention deletion trials – “Translation, re-insertion and swap [...] moves were chosen randomly...”. Furthermore, when explaining the moves used in NVT simulations (page 5), the authors also forgot to mention insertion and deletion trials – “...except that swap and CB/CFC moves were not used.”

Answer: In the RASPA convention, a “swap” move randomly attempts an insertion or a deletion with equal probability. We added an explanation of these swap moves on page 5.

6. A reference is missing for the Nosé-Hoover thermostat in page 5.

Answer: The citation has been added.

7. In Figure 1, the authors report a pore size distribution for MOF #667. They should explain how this was obtained, and which software was used for that purpose. Furthermore, it would be useful if the authors reported a set of common physical characteristics of the MOF, such as geometric surface area, pore volume, limiting pore diameter, etc.

Answer: The pore size distribution was obtained from Zeo++. A table of textual properties has been added to the SI as Table S6.

8. There is no validation of the molecular model used in the simulations. First of all, the authors should demonstrate that the models are able to predict adsorption in the chosen MOF, or (if experimental isotherms for that material are not available) at least in similar frameworks, either by carrying out additional simulations or by citing appropriate literature work. Second, and perhaps most importantly, the authors should discuss if the hysteresis behaviour observed in the simulations (namely a double-step in the isotherm) has been observed experimentally.

Without this kind of validation, one cannot be certain that the observed behaviour is not an artefact of the model.

Answer: In response to the reviewer's comment, we added a statement on page 5 of the revised manuscript that "The use of the UFF + TraPPE pair has been shown to provide good predictions of alkane adsorption in MOFs in previous work" along with several references. There is no experimental adsorption data available for this particular MOF, but making quantitative predictions for a particular material is not the focus of this work (as noted by Reviewer 2).

Double-step isotherms have been observed in various MOFs. For example, Struckhoff et al. (10.1002/admi.202000184) reported that argon in MIL-101(Cr) at 65 K exhibits a double-step in the adsorption and desorption isotherms, and there is a hysteresis loop. We have added a short description of this work on page 11 of the manuscript. However, experimental validation of the hysteresis loops is not the focus of this work. We focus on the exploration of hysteresis loops using a molecular model that captures the essential physics of the system and has been shown to predict adsorption properties in other MOFs in good agreement with experiment.

9. The study on finite-size effects (pages 14-17) is quite interesting and begs the question – what will happen in a realistic macroscopic sample of the MOF material. The authors offer a rather vague speculation in page 17, but I think this should be explored in more detail. From the data in Figure S9, it seems that the vdW loops near the centre of the hysteresis loop become smaller and closer to the equilibrium transition step. One may therefore argue that in a macroscopic material, those loops will become infinitesimally small so as to follow the real equilibrium curve (dashed blue line). However, the two (or perhaps more) vdW loops closest to the spinodal will remain as wider loops to connect to both metastable branches of the isotherm. This would be quite a distinct picture from the isotherm shown in Figure S9a for a single unit cell. Are the authors able to provide further information on the expected shape of the vdW loop as the system size tends to infinity? Is it possible to carry out an even larger simulation than 8 unit cells, or is that already at the edge of what is computationally feasible?

Answer: We thank the reviewer for this question. Using RASPA, calculations for methane in 8 unit cells required 2 months. If we used 16 unit cells, the time for the simulation would be 2^2 * (time for 8 unit cells), which would be 8 months. Although this is feasible in principle, we cannot finish such simulations within a reasonable time for the revision of this manuscript.

We agree with the reviewer that there are some interesting questions here and thank the reviewer for the careful observation and ideas about extrapolation of the vdW loop as the system size approaches infinity. From Figure 6 (previously Figure S9), we can see that as more unit cells are used, the vdW loops in the unstable region shrink, while the low-density spinodal with the lowest loading and the high-density spinodal with the highest loading quickly converge to certain fugacities. To support this claim, we extracted the fugacities of these two spinodals and plot them versus the number of unit cells used in (new) Figure S11. We can see that the fugacities of these two points converge as more unit cells are used.

Thus, we can anticipate that as the number of unit cells used approaches infinity, the vdW loops in the unstable region will approach the vertical binodal line while the two outmost spinodals remain constant and the metastable region depicted by the binodal and the spinodals will also remain. These points are now discussed on page 19 in the manuscript and in Figures S11 and S12 in the SI.

We thank the reviewer for the thoughtful comment again.

10. On a related point, it is not clear what happens to the vdW loop for the micropore as the system size increases. At the naked eye (Figure S9), it seems to disappear for the larger systems, but perhaps that is just due to the small scale of the figure. Can the authors examine the evolution of that loop in more detail, and again offer some interpretation of what should be expected in a realistic macroscopic sample?

Answer: A figure showing the zoomed-in canonical isotherms for the micropore has been added in the SI (Figure S9, the original Figure S9 was moved to the main text as Figure 6). We can see that the vdW loop for the micropore shrinks in size as more unit cells are used. Thus, we expect that for the micropore in a realistic macroscopic scenario, the vdW loop will shrink and approach the binodal line.

11. Perhaps the most interesting part of the paper, in my opinion, is the analysis of the effect of framework flexibility (Figure 10). However, the discussion of that section is underdeveloped. For example, the authors merely state that both the NVT and GCMC isotherms are shifted to the right by ~ 40 Pa, but do not offer a possible explanation for this shift. Perhaps by analysing the movement of the framework during adsorption in more detail, some additional insight can be gained? Also, it seems that the onset of the adsorption step in GCMC coincides with the equilibrium step determined from NVT. Is this a coincidence, or is there a physical reason why this should be the case?

Answer: We thank the reviewer for this question. The step in the GCMC isotherm and the binodal coincide because of the small size of the vdW loop. On pages 23 to 26 of the revised manuscript, we added several new paragraphs discussing the effect of framework flexibility, snapshots of methane during the simulation, the guest-guest and host-guest interaction energies, and the origin of this fugacity shift when framework flexibility is incorporated. The reference that the reviewer suggested in comment 1 was very helpful for our analysis. These suggestions and questions have improved this manuscript considerably, and we thank the reviewer again for the thoughtful comments.

Reviewer 2:

The study presented by Snurr and co-workers deals about a very interesting subject on the adsorption behavior within nanoporous frameworks having both micro- and mesopores. For this purpose, a complementary set of simulations are performed. I believe the results obtained by the authors are important and timely for the community working on adsorption characteristics of nanoporous frameworks and in general advise to publish the paper, albeit after taking into consideration a few suggestions.

I have a few questions and suggestions for improving the paper. Various suggestions intend to clarify the main message of the paper, making it more accessible for a broader audience and emphasizing more the new findings and physical chemistry insights.

1. The authors use several times (already in abstract) the nomenclature “in a metal-organic framework having both micropores and mesopores”. I think it would be better to introduce the rationale for the material more convincingly. If I understood correctly, the particular material is not crucial, it only needs to have clear micro- and mesopores and a few other characteristics which are mentioned by the authors. For this purpose, the authors took a material from their ToBaCCo database. It should be clearly emphasized that one particular material has been chosen with the proper motivation, but that the material as such is not the most important aspect. This is a fundamental study to obtain more insight into the adsorption properties of micro-/mesoporous materials.

Answer: We thank the reviewer for the kind suggestions. The abstract and the last paragraph of the Introduction have been modified accordingly.

2. I suggest to also make the main hypothesis/research result more clear in the abstract. Now it is mentioned “At low temperatures, the calculated isotherms exhibit sharp steps accompanied by hysteresis”. If I understood correctly, they found that both at low temperatures and for increasing molecule size the van der Waals loops grow. I thus suggest to also mention the influence of the molecule size in the abstract. In general the abstract would benefit from some rewriting to make the main conclusions and research hypothesis more clear. The introduction is well written and the references are appropriate. The only remark on page 3 is the introduction of the MOF, see my remark above.

Answer: Rather than putting some of this information in the Abstract, we have modified the Introduction and Conclusions sections of the paper.

Methods :

3. The authors used a series of force fields to perform the simulations. Can they comment on the dependence of the results on the particular choice of force field. Probably, for this study where it is the intention to give qualitative insights into the adsorption isotherms for hierarchical materials, the specific force field is no problem? Can the authors confirm or comment on this? It is well known that in general for the calculation of isotherms, the specific force field may be very important.

Answer: As the Reviewer points out, the specific force field isn't the focus of this paper, but the paper does discuss some details of force field, for example, the use of tail corrections, and their

impact on hysteresis and vdW loops. We can see from Figures 9 and 10 that tail corrections have a strong impact on some details of the hysteresis and vdW loops for methane and ethane adsorption in this MOF.

See also the response to comment #8 from Reviewer 1.

Results :

4. Overall, the results section is written in a technically very detailed way and the simulations are performed with care, however I invite the authors to emphasize their main physical chemistry insights and conclusions, in relation to the starting hypothesis more clearly throughout the paper.

Answer: We have modified the last paragraph of the Introduction and the Conclusions sections to emphasize the main conclusions of the work.

5. Figure 3 could maybe be improved, it is mentioned that the vdW loops overlap for the micro- and mesopores, but it would be interesting to indicate this more clearly on the figure.

Answer: In response to the reviewer's comment, we have colored the dots on the vdW loops differently for different pores and the overlapping region in Figure 3.

6. The supporting information is essential for following the paper. It might be considered to take up an additional figure in the main manuscript on the size effects, as I believe these are important conclusions. It could be Figure S9 possibly in adapted form to make is suited for the main manuscript.

Answer: Thanks for the comment. We moved Figure S9 to Figure 6 in the revised manuscript.

7. The sentence "The corresponding plots of grand potential versus loading are shown in Figure S10 and provide information on the free energy barriers between the different states" on page 16, is not fully clear. Can the authors comment more on the free energy barriers?

Answer: We changed Figure S10 to make the comparison more obvious and added a few lines discussing free energies on pages 18-19. As the unit cell size increases, the free energy barrier decreases.

Conclusions :

8. "In this work, we examined the adsorption of n-alkanes in a metal-organic framework, reference 41". It is not fully clear why reference 41 is given at this place.

Answer: We had cited Colon et al. here because we chose the MOF from their ToBaCCo database. However, we agree with the reviewer that this was not clear, so we have removed the reference from this sentence. (It is cited earlier in the paper, too.)

9. The conclusions found for the system size are very interesting, it is suggested that the canonical isotherms are the combination of many small vdW loops. Could the authors comment on the possible experimental validation of these conclusions? In how far are these conclusions realistic or can these only be observed in simulations?

Answer: As discussed in the response to Comment # 9 of Reviewer 1, we have added some additional discussion on how the canonical isotherm will look in a system with a macroscopic number of unit cells. We anticipate that the spinodals and vdW loops in the unstable region will converge to the binodal, while the low-density spinodal with the lowest loading (the most bottom right spinodal) and the high-density spinodal with the highest loading (the most top left spinodal) will remain at their converged locations. Please see Figure S11 for details. It might be possible to observe individual pores filling with ^{129}Xe NMR, but we have not added this to the paper, since it is somewhat speculative.

10. In the sentence “the results suggest that the canonical isotherm will be the combination of many small vdW loops – one for each pore”. Please specify the type of pore.

Answer: When we wrote “each pore” in this sentence, we meant each individual pore, not each “type of pore.” We have added the word “individual” in this sentence so that this is clearer.

11. Finally it might be interesting to give some perspective on the potential application importance of their findings. Nowadays many hierarchical materials are studied for particular applications and thus such perspective could increase the impact of the work.

Answer: To address this point, a sentence has been added to the Conclusions section of the manuscript: “The fundamental insights from this work may help improve adsorbents for applications such as adsorption cooling, atmospheric water harvesting, and carbon capture where it is often desirable to find conditions or materials that avoid hysteresis.”

Reviewer 3:

This paper applies GCMC/MD simulations in combination with the Widom insertion method to reveal the states of the adsorbed fluid inside the adsorption hysteresis loops. The particular novelty of the paper is the application of this analysis to the mesoporous metal organic frameworks (MOFs) and various hydrocarbons. To the best of my knowledge this is the first attempt to explore the metastable states and confined phase diagrams in MOFs.

MOFs is a relatively new class of materials and we are still in the process of understanding their fundamental adsorption characteristics. The current article seems to support the notion that mesoporous materials share fundamental physics of adsorption phenomena. On the other hand, MOFs seem to feature some new effects. For example, the increase of the number of loops in the system with the system size is quite interesting. From this perspective, I believe this paper will be of interest to a broad community of scientists working in the field of adsorption and adsorption in MOFs specifically.

The paper is clearly written, the methodology is well explained, and the references are appropriate. My recommendation is to publish this paper.

Few issues, where I think the authors could provide further comment are as follows:

1) Extrapolating the system size effects to large system, what would be the picture in this case? Do the authors expect the space inside the hysteresis loop filled in with infinite number of metastable states?

Answer: This important point was brought up by all 3 reviewers. Please see the response to Comment #9 from Reviewer 1.

2) Will the data and the simulation setups shared using some github depository system?

Answer: The data related to this work are all in the link below.

<https://figshare.com/articles/dataset/Data/21743354/2>

REVIEWERS' COMMENTS:

Reviewer #1 (Remarks to the Author):

The authors have provided a very detailed response to all reviewers' comments, and made several changes to improve their manuscript. I am happy to recommend this work for publication, and congratulate the authors on their work.

As a minor point for possible correction, I cannot see Figure S9 in the SI file. Perhaps it is just a glitch in the conversion to pdf, but worth checking.

Reviewer #2 (Remarks to the Author):

The authors have taken into account all comments of the reviewers, the manuscript is suitable for publication in its current form.